# The Forward and Lateral Tilt Angle of the Neck and Trunk Measured by Three-Dimensional Gait and Motion Analysis as a Candidate for a Severity Index in Patients with Parkinson's Disease

Hirofumi Matsumoto [1], Makoto Shiraishi [1], Ariaki Higashi [2], Sakae Hino [1], Mayumi Kaburagi [1], Heisuke Mizukami [1], Futaba Maki [1,3], Junji Yamauchi [4], Kenichiro Tanabe [4], Tomoo Sato [4] and Yoshihisa Yamano [1,4,*]

1   Department of Internal Medicine, Division of Neurology, St. Marianna University School of Medicine, Kawasaki 216-8511, Japan
2   Systemfriend Inc., Hiroshima 731-5125, Japan
3   Department of Neurology, Shin Yurigaoka General Hospital, Kawasaki 215-0026, Japan
4   Department of Rare Diseases Research, Institute of Medical Sciences, St. Marianna University School of Medicine, Kawasaki 216-8512, Japan
*   Correspondence: yyamano@marianna-u.ac.jp; Tel.: +81-44-977-8111

**Abstract:** (1) Objective: To evaluate the usefulness of a three-dimensional motion-analysis system (AKIRA®) as a quantitative measure of motor symptoms in patients with Parkinson's disease (PD). (2) Method: This study included 48 patients with PD. We measured their motion during 2 m of walking using AKIRA®, we calculated the tilt angles of the neck and trunk, ankle height, and gait speed, then we compared these parameters with the MDS-UPDRS and the Hoehn and Yahr scale. Furthermore, we measured these AKIRA indicators before and after 1 year of observation. (3) Results: The forward tilt angle of the neck showed a strong correlation with the scores on parts II, III, and the total MDS-UPDRS, and the tilt angle of the trunk showed a moderate correlation with those measures. The lateral tilt angle of the trunk showed a moderate correlation with a freezing of the gait and a postural instability. Regarding changes over the course of 1 year (*n* = 34), the total scores on part III of the MDS-UPDRS and the forward tilt angle of the neck improved, while the lateral tilt angle of the trunk worsened. (4) Conclusion: Taken together, the forward and lateral tilt angles of the neck and trunk as measured by AKIRA® can be a candidate for quantitative severity index in patients with PD.

**Keywords:** neck/trunk tilt angle; three-dimensional motion analysis; Parkinson's disease; MDS-UPDRS

## 1. Introduction

Parkinson's disease (PD) is a chronic, progressive neurological disorder, the prominent symptoms of which are motor symptoms consisting of resting tremor, akinesia, postural reflex disturbance, and rigidity. As these symptoms impair patients' motor functions [1], falls [2,3] and the freezing of the gait [3] occur frequently, reducing patients' QOL to a noticeable degree. Historically, the Hoehn and Yahr scale has been used to rate the severity of motor symptoms in PD patients [4]. Subsequently, the Unified Parkinson's Disease Rating Scale (UPDRS) was developed as a comprehensive evaluation index for various aspects of PD, including motor impairment and non-motor symptoms [5]. The recently revised version, known as the Movement Disorder Society UPDRS (MDS-UPDRS) [6], is widely used. While the MDS-UPDRS is widely used as a validated evaluation index, it may not detect small symptomatic changes, particularly those in motor symptoms, even when said changes are clear from gross observation, since it uses discontinuous numbers to rate PD symptoms in a stepwise manner. Therefore, the development of a scale with which to

evaluate motor symptoms quantitatively with continuous variables is in high demand for use not only in clinical practice but also in drug development.

As a result of recent, rapid progress being made in the development of various digital motion-analysis devices, an increasing number of reports have accumulated, describing the utilization of such devices for quantitative motion analysis in PD patients [7]. In particular, since three-dimensional analysis is essential for the accurate evaluation of motor symptoms in PD patients, the analysis systems for such a purpose require multiple cameras and sensors or markers which must be attached to patients during data acquisition. Such requirements represent major obstacles to the introduction of the systems in clinical settings [7]. AKIRA® (https://www.systemfriend.co.jp/en/service/akira accessed on 10 June 2022) combines video and infrared sensors for the quantitative, three-dimensional evaluation of body movements; with a single camera and no markers attached to the patient's body, the system videorecords body movements, automatically recognizes major joint points in the whole body, and records the real-time locations of these points as absolute coordinate values in the horizontal (*X*-axis), vertical (*Y*-axis), and anteroposterior (*Z*-axis) directions with the camera as the origin [8,9].

The observation of body motion is critical for the evaluation of PD, as bradykinesia is an essential finding in the Movement Disorder Society Clinical Diagnostic Criteria for Parkinson's disease [10]. Another characteristic of PD is postural abnormalities, such as a chin-on-chest or head-forward posture and Pisa syndrome. PD specialists routinely make their ratings of the severity of patients' conditions based on body movements and the level of postural abnormalities demonstrated while the patient enters the examination room; however, no quantitative, easy-to-use evaluation methods have been established for body movements and postural abnormalities. Therefore, in this study, we conducted three-dimensional gait and motion analysis using AKIRA® in PD patients to measure the tilt angles of the neck and trunk during walking, and we compared these parameters with the MDS-UPDRS to evaluate their usefulness as measures by which to rate the severity of PD.

## 2. Materials and Methods

### 2.1. Participants

This was a prospective, observational study of PD patients. Patients with PD who visited the St. Marianna University Hospital between September 2020 and March 2022 and who met the following inclusion criteria were included in this study. Inclusion criteria were: (1) patients with a definitive diagnosis of PD based on the Movement Disorder Society Clinical Diagnostic Criteria for Parkinson's disease [10]; (2) patients aged ≥20 years and ≤80 years; (3) patients who had had no changes in medication within 1 week of evaluation; (4) for patients experiencing the wearing-off phenomenon, the "on" status should have been confirmed during evaluation; (5) patients who were capable of understanding instructions; (6) a Hoehn and Yahr rating of II–IV; (7) patients without psychological symptoms interfering with daily life, such as severe hallucinations; and (8) patients capable of maintaining a standing posture. Exclusion criteria were: (1) patients with PD-related diseases other than PD; (2) patients with severely limited passive range of motion; (3) patients who had experienced a rapid exacerbation of PD symptoms within 1 week; and (4) patients who had developed complications interfering with motor functions within 1 month, such as fractures. Patients with an observation duration of 1 year after enrollment underwent the analysis at the time of enrollment and 1 year after enrollment, and they had no restrictions on their treatment details during the period. This study was conducted with approval from the clinical study committee of the St. Marianna University School of Medicine (approval number: 4885) and written consent from the patients.

### 2.2. Clinical Endpoints

The following pieces of patient information were collected at the time of enrollment: age, sex, BMI, and duration of illness. At the time of enrollment and 12 months (±2 months)

thereafter, the participants underwent investigations assessing their L-dopa equivalent daily dose (LEDD) [11,12], Hoehn and Yahr scale, and MDS-UPDRS, as well as videorecording of their gait motion with AKIRA®. The MDS-UPDRS consists of four parts from I to IV: Part I: Non-motor aspects of experiences of daily living (nM-EDL); Part II: Motor aspects of experiences of daily living (M-EDL); Part III: Motor examination; and Part IV: Motor complications [6] (see Supplementary Table S1). For the MDS-UPDRS items that ask about the left/right and upper/lower limbs independently, the total scores for all limbs were used for evaluation (see Supplementary Table S1).

### 2.3. Gait Assessment by AKIRA®

#### 2.3.1. Camera Placement and Videorecording

We used a high-performance video camera equipped with an infrared device (Kinect v2), which is an infrared light source, and which is sensitive to infrared light [8]. The Kinect v2 is a range-imaging camera system employing time-of-flight techniques to resolve the distance between the camera and the subject for each point of the image by measuring the round-trip time of an infrared light signal emitted by the camera. The camera was horizontally placed at a distance of 4 m from the subject's position and adjusted to a height of 80 cm from the floor. Before beginning to walk, subjects remained in a sitting position to rest for 5 min and were confirmed as having no fatigue. Then, the subjects were instructed to walk naturally toward the camera, and gait motions from the 4 m point to the 2 m point were recorded every 1/30 s. The subjects were instructed to pass the 2-meter point without stopping. The subjects maintained their faces toward the camera, and the horizontal, vertical, and anteroposterior directions were defined as the *X*-axis, *Y*-axis, and *Z*-axis, with the right, upper, and front sides being the + direction of each respective axis.

#### 2.3.2. Data Collection

In this study, virtual markers were placed at 5 points (head, neck, spine base, ankle left, and ankle right) using AKIRA® [9] (Figure 1), and (a) tilt angle, (b) ankle height, and (c) walking speed over a distance of 2 m from the starting point to the goal were calculated.

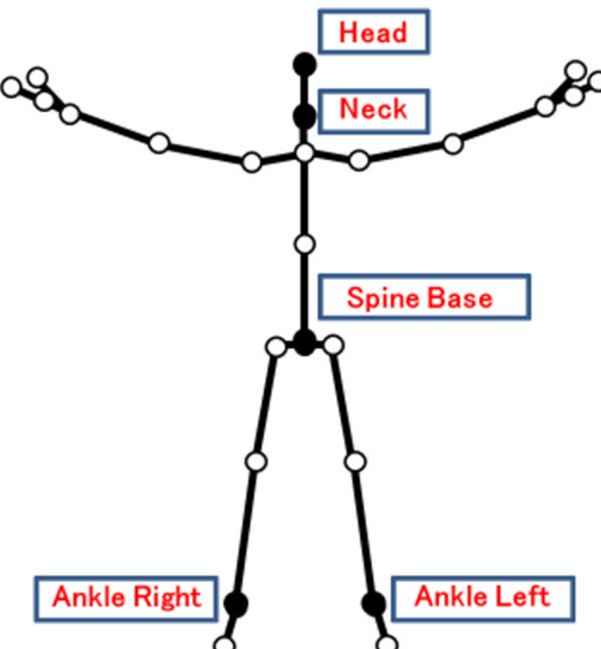

**Figure 1.** Virtual markers measured by AKIRA®.

Of the 25 virtual markers that can be measured by AKIRA®, 5 were selected for this study.

Tilt Angle: lateral tilt angles between the head and neck and between the neck and spine base were calculated from the (*X*, *Y*) coordinates of each set of the two virtual markers using the Atan2 function [9]. Forward tilt angles between the head and neck and between the neck and spine base were calculated from the (*Y*, *Z*) coordinates of each set of the two virtual markers using the Atan2 function. Mean, maximum, and minimum values were calculated for the forward/lateral tilt angles of the neck and the trunk as determined every 1/30 s.

Ankle Height: The amplitude difference between the ankle left and ankle right heights (*Y*) was defined as Ankle R-L.

Walking speed: The speed was calculated from the distance moved on the *Z*-axis (m) at the spine base and the measurement time (seconds, $1/30\times$ number of frames recorded).

*2.4. Statistical Analysis*

Spearman's rank correlation coefficient was used to measure the relationships between the tilt angles, ankle heights, and walking speeds with the MDS-UPDRS item scores, the Parts I, II, III, and IV subtotal scores, the total score, and the Hoehn and Yahr scale. The tightness of the correlations were classified into the following 4 groups based on the correlation coefficient: strong, $\geq 0.6$; moderate, $\geq 0.4$ and $< 0.6$; weak, $\geq 0.2$ and $< 0.4$; and none, $< 0.2$. The Wilcoxon signed rank test was used to detect changes in each variable over 1 year from the baseline. SPSS ver. 25 (IBM SPSS Statistics for Windows, IBM Corp., Armonk, NY, USA) was used for all statistical analyses. Numerical values are shown in the form of mean (median) or mean $\pm$ SD. All statistical tests were two-tailed tests with a significance level of $p < 0.05$.

**3. Results**

A total of 60 patients were enrolled between September 2020 and March 2022. (These patients are referred to as "study participants".) As shown in Figure 2, among the study participants, 4 were found ineligible after enrollment, and 8 were excluded because of incomplete imaging; thus, 48 PD patients underwent motion analyses with AKIRA®. (These patients are referred to as "eligible patients".) Of the eligible patients, 34 PD patients underwent the usual treatment for 1 year. (These patients are referred to as "traceable participants") (see Figure 2).

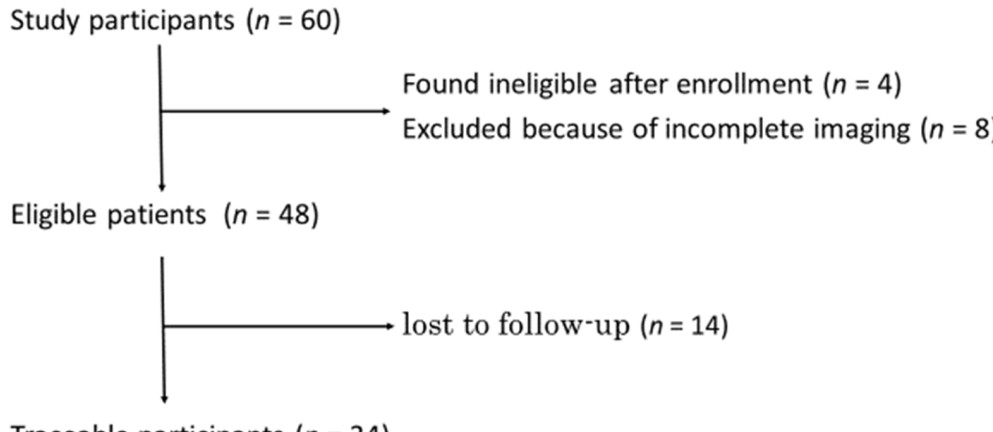

**Figure 2.** Flow chart of enrolment.

The background data for the 48 eligible patients were: 26 men and 22 women; age, $69.0 \pm 8.6$ years; duration of illness, $7.5 \pm 5.2$ years; BMI, $22.6 \pm 3.8$ kg/m$^2$; and Hoehn and Yahr scale, $2.6 \pm 0.74$ (Table 1).

**Table 1.** Patient characteristics (*n* = 48).

| Characteristic | | Values * |
|---|---|---|
| Age (years) | | 69.0 ± 8.6 |
| Number of men | | 26 (54.2%) |
| BMI | | 22.6 ± 3.8 |
| Duration of illness (years) | | 7.5 ± 5.2 |
| Hoehn and Yahr scale | | 2.6 ± 0.74 |
| MDS-UPDRS | Part I | 10.1 ± 5.8 |
| | Part II | 13.7 ± 8.4 |
| | Part III | 27.4 ± 15.3 |
| | Part IV | 4.3 ± 4.5 |
| | total | 55 ± 29 |
| LEDD (mg) | | 653 ± 521 |
| Levodopa/DCI (mg) | | 426 ± 393 |
| Dopamine Agonist | | 23 (47.9%) |
| Ropinirole | | 8 (16.7%) |
| Rotigotine | | 11 (22.9%) |
| Pramipexole | | 5 (10.4%) |
| MAO-B inhibitors | | 17 (35.4%) |
| Selegirine | | 4 (8.3%) |
| Rasagirine | | 8 (16.7%) |
| Safinamide | | 5 (10.4%) |
| COMT inhibitors | | 10 (20.8%) |
| Entacapone | | 9 (18.8%) |
| Opicapone | | 1 (2.1%) |
| Others | | |
| Amantadine | | 4 (8.3%) |
| Istradefylline | | 10 (20.8%) |
| Zonisamide | | 11 (22.9%) |
| Trihexyphenidyl | | 2 (4.2%) |

* Mean ± standard deviation, number (proportion). Legend: BMI = Body Mass Index; UPDRS = Unified Parkinson's Disease Rating Scale; LEDD = L-dopa equivalent daily dose; DCI = dopa decarboxylase inhibitor; MAO-B = monoamine oxidase B; COMT = catechol-O-methyl transferase.

### 3.1. Reliability of Measurement Data

First, to test the reliability of the measurements with AKIRA®, the gait motion data were acquired 3 consecutive times with AKIRA® in 12 of the 48 eligible patients who consented to multiple measurements, and coefficient of variation (CV) values for different parameters were calculated (Table 2). The %CV was ≤25 for mean and maximum forward and lateral tilt angles of the neck and trunk; however, the %CV was larger for the minimum forward and lateral tilt angles of the neck and trunk. The %CV was also ≤25 for Ankle R-L and speed. Based on these results, we concluded that the reliability was good for all measures except for the minimum forward and lateral tilt angles of the neck and trunk, and in subsequent analyses, the mean and maximum forward and lateral tilt angles of the neck and trunk were measured, and the number of video recordings taken for each measurement was 1.

**Table 2.** Coefficient of variation (CV) values for different parameters (*n* = 12).

| Index | Neck, Forward | | | Neck, Lateral | | | Trunk, Forward | | | Trunk, Lateral | | | Angle R-L | Speed |
|-------|-----|-----|-----|-----|-----|-----|-----|-----|-----|-----|-----|-----|-----|-----|
| | Ave | Max | Min | Ave | Max | Min | Ave | Max | Min | Ave | Max | Min | | |
| %CV | 6 | 9 | 93 | 16 | 25 | 81 | 13 | 14 | 71 | 22 | 12 | 49 | 23 | 16 |

Legend: %CV = percent coefficient of variation.

### 3.2. Correlation between AKIRA® Measurements and Clinical Parameters

Next, correlations between tilt angles measured with AKIRA® and the MDS-UPDRS and the Hoehn and Yahr scale were analyzed in 48 PD patients (Table 3).

**Table 3.** Relationships between the neck/trunk tilt angles measured by AKIRA® during walking and the MDS-UPDRS and the Hoehn and Yahr scale.

| | Neck, Forward | | Neck, Lateral | | Trunk, Forward | | Trunk, Lateral | | Ankle | |
|-------|-----|-----|-----|-----|-----|-----|-----|-----|-----|-----|
| | Ave | Max | Ave | Max | Ave | Max | Ave | Max | R-L | Speed |
| 1.1 | | | | | | | | | | |
| 1.2 | 0.269 | 0.269 | | | | 0.220 | | 0.335 | | |
| 1.3 | 0.243 | 0.228 | | | | | | 0.263 | | |
| 1.4 | 0.282 | 0.255 | | | | | | | | |
| 1.5 | 0.219 | 0.248 | | | | | | | | |
| 1.6 | | | | | | | | | | 0.299 |
| 1.7 | 0.318 | 0.228 | | | | | | | | |
| 1.8 | 0.334 | 0.322 | | | | | | | | |
| 1.9 | | | | | | | | | | |
| 1.10 | 0.263 | | | | 0.200 | | | | | |
| 1.11 | 0.345 | 0.271 | | | | | | | | |
| 1.12 | | | | | | | | | | |
| 1.13 | 0.244 | 0.293 | | | | | | | | |
| I Total | 0.448 | 0.395 | | | | | | 0.223 | | |
| 2.1 | 0.413 | 0.358 | | | | | | | | |
| 2.2 | 0.273 | 0.216 | | | | | | | | 0.251 |
| 2.3 | 0.303 | 0.377 | | | 0.242 | 0.295 | | | | |
| 2.4 | 0.496 | 0.450 | | | 0.315 | 0.359 | 0.290 | 0.396 | | |
| 2.5 | 0.538 | 0.511 | | | 0.264 | 0.295 | | | | |
| 2.6 | 0.452 | 0.406 | | | 0.261 | 0.302 | | | | |
| 2.7 | 0.415 | 0.371 | | | | | | | | |
| 2.8 | 0.381 | 0.376 | | | 0.202 | 0.216 | | 0.284 | | |
| 2.9 | 0.368 | 0.260 | | | | | | | | |
| 2.10 | | | | | | | | | | |
| 2.11 | 0.488 | 0.464 | 0.326 | 0.312 | 0.225 | 0.267 | | 0.283 | | |
| 2.12 | 0.381 | 0.366 | | | | | | | | |
| 2.13 | 0.438 | 0.417 | 0.294 | 0.277 | 0.451 | 0.524 | 0.229 | 0.432 | | |
| II Total | 0.619 | 0.581 | | | 0.396 | 0.431 | | 0.357 | | |
| 3.1 | 0.231 | 0.208 | | | 0.303 | 0.266 | | | | |
| 3.2 | 0.335 | 0.301 | | | 0.232 | 0.231 | | | | 0.209 |
| 3.3 | 0.376 | 0.399 | 0.200 | | | | | | | |
| 3.4 | 0.483 | 0.470 | | | 0.330 | 0.330 | | | | |
| 3.5 | 0.334 | 0.454 | | | | 0.250 | | | | |

**Table 3.** *Cont.*

| | Neck, Forward | | Neck, Lateral | | Trunk, Forward | | Trunk, Lateral | | Ankle | |
| --- | --- | --- | --- | --- | --- | --- | --- | --- | --- | --- |
| | Ave | Max | Ave | Max | Ave | Max | Ave | Max | R-L | Speed |
| 3.6 | 0.357 | 0.379 | | | | | | 0.225 | | |
| 3.7 | 0.389 | 0.315 | 0.238 | | 0.273 | 0.247 | 0.211 | 0.361 | | |
| 3.8 | 0.495 | 0.440 | | | 0.364 | 0.331 | | | | |
| 3.9 | 0.511 | 0.497 | | | 0.249 | 0.293 | | | | |
| 3.10 | 0.396 | 0.474 | | 0.233 | | 0.233 | | 0.297 | | |
| 3.11 | 0.283 | 0.261 | | | | 0.241 | | | | |
| 3.12 | 0.482 | 0.556 | 0.222 | | 0.433 | 0.531 | | 0.408 | | |
| 3.13 | 0.296 | 0.275 | | | | 0.215 | | | | |
| 3.14 | 0.406 | 0.382 | | | 0.273 | 0.358 | 0.209 | 0.391 | | |
| 3.15 | 0.221 | 0.279 | | | | | | | | |
| 3.16 | 0.218 | 0.338 | | 0.221 | 0.264 | 0.240 | | 0.235 | | |
| 3.17 | | 0.204 | | | | | | | | |
| 3.18 | | 0.220 | 0.215 | | | | | | | |
| III Total | 0.602 | 0.648 | | | 0.391 | 0.418 | | 0.332 | | |
| 4.1 | | | | | | | | | | |
| 4.2 | | | | | | | | | | |
| 4.3 | 0.293 | | | | 0.402 | 0.427 | 0.363 | 0.379 | | |
| 4.4 | 0.365 | 0.247 | | | 0.283 | 0.313 | | | | |
| 4.5 | 0.247 | | | | 0.241 | 0.224 | | | | |
| 4.6 | | | | | | | | | | |
| IV Total | 0.300 | | | | 0.247 | 0.264 | | | | |
| Total | 0.635 | 0.610 | | | 0.419 | 0.439 | | 0.344 | | |
| H&Y | 0.385 | 0.412 | 0.282 | 0.329 | 0.344 | 0.439 | | 0.333 | | |
| | $r_s < 0.2$ | | | $0.2 \leq r_s < 0.4$ | | | $0.4 \leq r_s < 0.6$ | | | $0.6 \leq r_s$ |

The forward tilt angles of the neck (mean and maximum) showed strong correlations with scores for various items in parts II and III of the MDS-UPDRS; of these, the correlations of the mean forward tilt angle of the neck with scores for parts II and III and the total score of the MDS-UPDRS were particularly strong ($r_s$ = 0.619, 0.602, 0.635, respectively) (Table 3).

The maximum forward tilt angle of the trunk also had moderate correlations with the parts II, III, and total scores of the MDS-UPDRS ($r_s$ = 0.431, 0.418, 0.439, respectively). The maximum lateral tilt angle of the trunk correlated moderately with the scores for Freezing (2.13 in part II of the MDS-UPDRS) and Postural Stability (3.12 in part III of the MDS-UPDRS) ($r_s$ = 0.432, 0.408, respectively) (Table 3).

The Hoehn and Yahr scale showed moderate correlations with the maximum forward tilt angles of the neck and trunk ($r_s$ = 0.412, 0.439, respectively) (Table 3). The Ankle R-L and walking speed had few correlations with either the MDS-UPDRS or the Hoehn and Yahr scale.

The relationships between the tilt angles, ankle heights, and walking speeds measured by AKIRA® with the item scores on Parts I, II, III, and the Part IV subtotal scores, and the total scores of the MDS-UPDRS and the Hoehn and Yahr scale in 48 PD patients are shown. Different colors denote the categories of the correlation coefficient: $r_s < 0.2$; $0.2 \leq r_s < 0.4$; $0.4 \leq r_s < 0.6$; and $0.6 \leq r_s$. MDS-UPDRS: Movement Disorder Society (MDS)-sponsored revision of the Unified Parkinson's Disease Rating Scale; H&Y: Hoehn and Yahr.

### 3.3. Time Course of Each Indicator

To determine whether various measurements with AKIRA® are useful indicators for evaluating spontaneous and treatment-induced changes over time in motor functions in the real world, changes in the MDS-UPDRS and AKIRA® measurements (forward tilt angle of

the neck, forward tilt angle of the trunk, and lateral tilt angle of the trunk) were monitored for 1 year in 34 traceable participants. Note that the LEDD (L-dopa equivalent daily dose) increased significantly from 649 mg to 722 mg over the course of the 1-year observation period ($p = 0.002$ by the Wilcoxon signed-rank test; data not shown).

As shown in Figure 3, the subtotal score for Part III of the MDS-UPDRS after 1 year was significantly improved compared to the baseline score ($p = 0.020$); however, the Part II subtotal and the total scores of the MDS-UPDRS and the Hoehn and Yahr scale remained unchanged, and the Part I subtotal score of the MDS-UPDRS worsened significantly ($p = 0.047$). Meanwhile, for the AKIRA® measurements, the mean and maximum forward tilt angles of the neck were significantly improved ($p = 0.013, 0.003$, respectively); however, the mean and maximum forward tilt angles of the trunk showed no significant changes. Both the mean and maximum lateral tilt angles of the trunk worsened significantly ($p = 0.008$, $0.021$, respectively).

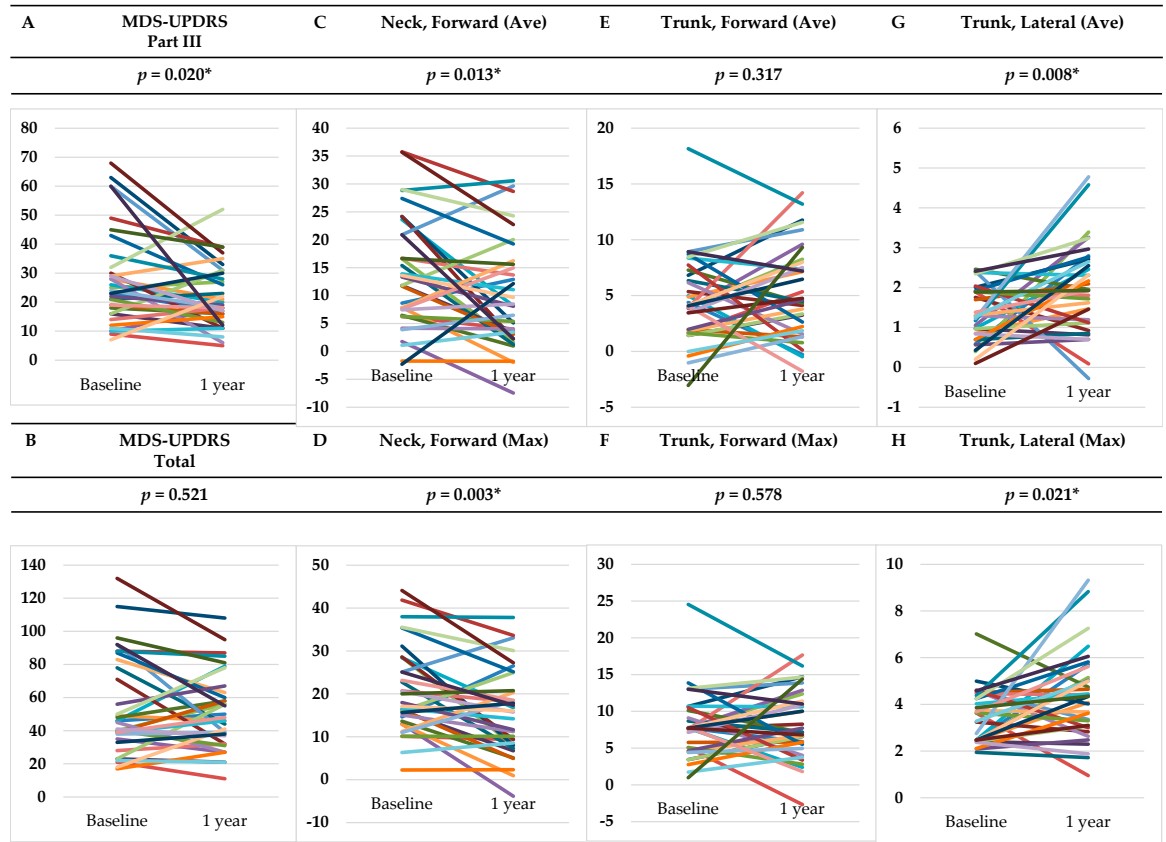

**Figure 3.** Changes over time in MDS-UPDRS Part III/Total score and tilt angles: (**A**) MDS-UPDRS Part III subtotal score; (**B**) MDS-UPDRS total score; (**C**) mean forward tilt angle of the neck; (**D**) maximum forward tilt angle of the neck; (**E**) mean forward tilt angle of the trunk; (**F**) maximum forward tilt angle of the trunk; (**G**) mean lateral tilt angle of the trunk; and (**H**) maximum lateral tilt angle of the trunk. * $p < 0.05$.

## 4. Discussion

AKIRA®, which does not require markers to be attached to the body, has been used to measure changes in the balance and various joint angles during walking in healthy volunteers [8,9], but it has not been used very commonly in clinical settings. Thus, we were initially concerned about data reproducibility as a possible problem. In this study, %CV values for the maximum and mean forward and lateral tilt angles of the neck and trunk measured with AKIRA® were found to range from 6 to 25%, showing relatively good reliability levels (Table 2) although the minimum angles had poor reproducibility. These

results demonstrate that the maximum and mean angles can be used for the evaluation of motor functions in PD patients.

The correlation between walking speed and MDS-UPDRS was unexpectedly weak (Table 3). In fact, gait performance has been reported to correlate with only some Part-III elements of the MDS-UPDRS, rather than the whole MDS-UPDRS subgroup [13,14], and our results were not contradictory.

In this study, the forward tilt angles of the neck and trunk were shown to correlate with various MDS-UPDRS items, and this finding suggests that the forward tilt angles of the neck and trunk are more important than the walking speed for rating the severity of PD in patients. In considering the correlation between the forward tilt angles and the severity in PD patients, the concept of higher-level gait disorders proposed by Nutt JG et al. to describe gait disorders in elderly people, which are not explainable by simple gait-element disorders, is of interest [15]. Higher-level gait disorders include the concept of cautious gait, which refers to walking in an overcautious manner [16]. The forward tilt angles reflected the severity of various symptoms more than the simple walking speed, presumably because cautious gait also became more noticeable in PD patients as the severity increased.

Meanwhile, the lateral tilt angle of the trunk correlated with fewer MDS-UPDRS items than did the forward tilt angle, as well as with some MDS-UPDRS subscales, such as 2.13 Freezing and 3.12 Postural stability (Table 3). Among PD patients, the Non-Motor Symptoms Scale (NMSS) scores of patients with Pisa syndrome characterized by severe lateral flexion has been reported as being significantly higher than that of patients without Pisa syndrome [17]. Therefore, the lateral tilt angle is likely to also reflect factors other than motor impairment in PD patients.

Furthermore, we analyzed changes over time in the AKIRA® measurements in this study. As shown in Figure 3, scores on the MDS-UPDRS Part III were significantly improved after 1 year of conventional treatment, suggesting that the objective motor disorders had improved; however, the Part I scores reflecting non-motor symptoms worsened. In this population of PD patients, the forward tilt angle of the neck was improved significantly, while the lateral tilt angle of the trunk worsened significantly. These findings suggest that the forward tilt angle of the neck reflects improvements in motor disorders, and the lateral tilt angle of the trunk reflects changes in non-motor symptoms. These results also suggest that, although conventional drug therapy improves motor symptoms, it is ineffective in improving the lateral tilt angle of the trunk. Moving forward, the development of a treatment strategy to improve the lateral tilt angle of the trunk is considered to be important for improving the QOL of PD patients. AKIRA® also appears to be useful in the evaluation of PD patients for this purpose, as it can quantitatively measure the lateral tilt angle of the trunk.

A limitation of this study is that it included only a single patient population from a single institution, so there has been no validation regarding whether similar results would be obtained in different patient populations. Multi-center studies are desirable.

## 5. Conclusions

The values measured with AKIRA® were confirmed to be quantitative and reliable. Therefore, the forward and lateral tilt angles of the neck and trunk determined as by three-dimensional gait and motion analysis are considered to be candidate measures for severity evaluation in PD patients. Moving forward, multi-center prospective studies may be necessary for verification.

**Supplementary Materials:** The following supporting information can be downloaded at: https://www.mdpi.com/article/10.3390/neurolint14030061/s1, Table S1: MDS-UPDRS Score Sheet.

**Author Contributions:** Conceptualization, H.M. (Hirofumi Matsumoto), M.S. and Y.Y.; data curation, H.M. (Hirofumi Matsumoto), M.S., S.H., M.K., H.M. (Heisuke Mizukami) and F.M.; formal analysis, H.M. (Hirofumi Matsumoto), M.S. and K.T.; funding acquisition, Y.Y.; Investigation, H.M. (Hirofumi Matsumoto), M.S., A.H., S.H., M.K., H.M. (Heisuke Mizukami), F.M., J.Y., K.T., T.S. and

Y.Y.; methodology, H.M. (Hirofumi Matsumoto), M.S. and J.Y.; project administration, Y.Y.; software, A.H.; supervision, M.S. and Y.Y.; validation, H.M. (Hirofumi Matsumoto) and K.T.; visualization, H.M. (Hirofumi Matsumoto); writing—original draft, H.M. (Hirofumi Matsumoto) and M.S.; writing—review and editing, A.H., S.H., M.K., H.M. (Heisuke Mizukami), F.M., J.Y., K.T., T.S. and Y.Y. All authors have read and agreed to the published version of the manuscript.

**Funding:** This work was supported by grants from the Practical Research Project for Rare/Intractable Diseases of the Japan Agency for Medical Research and Development (AMED, No. JP22ek0109529h, JP22ek0109441h, JP22ek0109493s, and JP22ek0109548s), Rare and Intractable Diseases from the Ministry of Health, Labour and Welfare of Japan (No. JPMH22FC1013); and the Japan Society for the Promotion of Science (JSPS) KAKENHI (No. JP22H02987).

**Institutional Review Board Statement:** This study was performed in line with the principles of the Declaration of Helsinki. This study was approved by the Ethics Committee of the St. Marianna University School of Medicine (approval No. 4885, 17 September 2020).

**Informed Consent Statement:** Informed consent was obtained from all subjects involved in the study.

**Data Availability Statement:** The data are available upon request.

**Acknowledgments:** The authors thank Junya Ono, Ryo Nakagawa, Kiwa Warashina, Sarie Mizuno, and Shinobu Tochimoto at the Department of Rehabilitation, St. Marianna University Hospital.

**Conflicts of Interest:** The authors declare that there are no conflict of interest regarding this article.

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
