# Peer review of "The Forward and Lateral Tilt Angle of the Neck and Trunk Measured by Three-Dimensional Gait and Motion Analysis as a Candidate for a Severity Index in Patients with Parkinson’s Disease"

_2035-8377, doi:10.3390/neurolint14030061_

Round 1

Reviewer 1 Report

Thank you for providing a very interesting new concept. Abnormalities in posture of Parkinson's disease patients have been described ever since the disease was identified, but a quantitative measurement of neck tilt angle has not been previously rigorously studied. This manuscript will spark important conversations about the severity of neck tilt as it relates to overall disease severity. 

Author Response

Thank you for your positive comment. We are honored to have your feedback and appreciate your encouraging suggestion.

Reviewer 2 Report

Motor assessment in an important issue in evaluating diseases progression or the efficacy of therapy in patients with Parkinson’s disease. Based on the progresses of imaging capture, image processing and analyses, quantified evaluation of postures and gaits become popular currently. The authors used AKIRA® system for three-dimensional evaluation of body movements of patients with Parkinson’s diseases.  

The followings are my comments and suggestions:

Major:

1.        Method 2.3.2 Data collection: According to the description in section 2.3.1, it seems that there is only one camera facing toward the subject to collect images, then how does the distance in the z-axis is derived? Although the authors have cited a reference to valid the reliability of AKIRA® in assessing the trunk, hip and knee angle during walking on a flatland and a treadmill. However, from my point of view, it is difficult to figure out how to estimate the gait speed and assess forward tilt angle of neck and trunk during motion by a camera in front of the patients. Providing a real image that shows how the distance in x, y, and z directions are calculated will be helpful to understand.

2.        Result 3.1 reproducibility of measurement data. The authors used coefficient of variation to estimate the reproducibility of the motion parameters in 12 out of 48 patients, who received three consecutive times of assessment with AKIRA®. Generally, this method is used to assess the diversity of the data instead of reproducibility, especially in the Parkinson’s disease patients with diverse clinical presentations. Meanwhile, presentation of CV in table 2 is incorrect either. The authors should change their method to evaluate the reproducibility [1].

3.        Figure 4. The statistical method is not clear the authors should elucidate the methodology to assess the correlation between the indicators and L-dopa-equivalent daily dose in details. 

Minor

1.        Section 2.3.1, Line 1: What does it mean by 'equipped with an infrared device'? Is it an infrared light source or the camera is sensitive to infrared light?

Reference

[1]   K. V. Bunting, R. P. Steeds, L. T. Slater, J. K. Rogers, G. V. Gkoutos, and D. Kotecha, "A Practical Guide to Assess the Reproducibility of Echocardiographic Measurements," Journal of the American Society of Echocardiography. 2019;32:1505-15.

Author Response

Thank you for your comments. We have revised our manuscript to reflect your suggestions and hope you agree with our revision. Below are the point-by-point responses to your comments.

Major comment #1

Method 2.3.2 Data collection: According to the description in section 2.3.1, it seems that there is only one camera facing toward the subject to collect images, then how does the distance in the z-axis is derived? Although the authors have cited a reference to valid the reliability of AKIRA® in assessing the trunk, hip and knee angle during walking on a flatland and a treadmill. However, from my point of view, it is difficult to figure out how to estimate the gait speed and assess forward tilt angle of neck and trunk during motion by a camera in front of the patients. Providing a real image that shows how the distance in x, y, and z directions are calculated will be helpful to understand.

Thank you for your helpful comments. AKIRA is using a Kinect v2 (which is well known to be used by X-box). Kinect v2 is a so-called Time-of -flight camera, which is a range imaging camera system employing time-of-flight techniques to resolve distance between the camera and the subject for each point of the image, by measuring the round trip time of an infrared light signal provided by a camera. According to your suggestion, we added the following sentence in Method 2.3.1. in the revised manuscript.

“We used a high-performance video camera equipped with an infrared device (Kinect v2), which is an infrared light source and is sensitive to infrared light [8]. The Kinect v2 is a range imaging camera system employing time-of-flight techniques to resolve distance between the camera and the subject for each point of the image, by measuring the round trip time of an infrared light signal provided by a camera.”

Major comment #2

Result 3.1 reproducibility of measurement data. The authors used coefficient of variation to estimate the reproducibility of the motion parameters in 12 out of 48 patients, who received three consecutive times of assessment with AKIRA®. Generally, this method is used to assess the diversity of the data instead of reproducibility, especially in the Parkinson’s disease patients with diverse clinical presentations. Meanwhile, presentation of CV in table 2 is incorrect either. The authors should change their method to evaluate the reproducibility [1].

Thank you for your comments. Based on your comments and reference, we realized that we analyzed “Reliability” but not “Reproducibility”. As you suggest, it would be ideal to analyze “Reproducibility”. However, it requires complex computation which is difficult for us. Therefore, according to your comments, we replaced from “Reproducibility” to “Reliability” throughout the revised manuscript.

Major comment #3

Figure 4. The statistical method is not clear the authors should elucidate the methodology to assess the correlation between the indicators and L-dopa-equivalent daily dose in details. 

Sorry for the confusion. Unfortunately, we did not analyze the correlation between the indicators and L-dopa-equivalent daily dose (LEDD) because optimal effective doses are quite different among patients. In Result 3.3, we only analyzed the change of LEDD before and after the one-year observation. To make this point clear, we revised the Result 3.3 as follows.

LEDD (L-dopa-equivalent daily dose) increased significantly from 649 mg to 722 mg before and after one-year observation (p = 0.002 by the Wilcoxon signed rank test, data not shown).

Minor comment #1

Section 2.3.1, Line 1: What does it mean by 'equipped with an infrared device'? Is it an infrared light source or the camera is sensitive to infrared light?

Thank you for your comments. As we answered to your major comment #1, the camera is an infrared light source and is sensitive to infrared light. According to your suggestion, we added the following sentence in Method 2.3.1. in the revised manuscript.

“We used a high-performance video camera equipped with an infrared device (Kinect v2), which is an infrared light source and is sensitive to infrared light [8]. The Kinect v2 is a range imaging camera system employing time-of-flight techniques to resolve distance between the camera and the subject for each point of the image, by measuring the round trip time of an infrared light signal provided by a camera.”

Round 2

Reviewer 2 Report

Nil